# NMR Structure Determinations of Small Proteins Using Only One Fractionally 20% ^13^C- and Uniformly 100% ^15^N-Labeled Sample

**DOI:** 10.3390/molecules26030747

**Published:** 2021-02-01

**Authors:** Harri A. Heikkinen, Sofia M. Backlund, Hideo Iwaï

**Affiliations:** Institute of Biotechnology, University of Helsinki. P.O. Box 65, FIN-00014 Helsinki, Finland; harri.a.heikkinen@helsinki.fi (H.A.H.); sofia.backlund@helsinki.fi (S.M.B.)

**Keywords:** NMR structure determination, fractional ^13^C-labeling, biosynthetically directed fractional ^13^C-labeling, NMR, biosynthetic pathways, protein structure, CBM64, SpaC, NMR assignment

## Abstract

Uniformly ^13^C- and ^15^N-labeled samples ensure fast and reliable nuclear magnetic resonance (NMR) assignments of proteins and are commonly used for structure elucidation by NMR. However, the preparation of uniformly labeled samples is a labor-intensive and expensive step. Reducing the portion of ^13^C-labeled glucose by a factor of five using a fractional 20% ^13^C- and 100% ^15^N-labeling scheme could lower the total chemical costs, yet retaining sufficient structural information of uniformly [^13^C, ^15^N]-labeled sample as a result of the improved sensitivity of NMR instruments. Moreover, fractional ^13^C-labeling can facilitate reliable resonance assignments of sidechains because of the biosynthetic pathways of each amino-acid. Preparation of only one [20% ^13^C, 100% ^15^N]-labeled sample for small proteins (<15 kDa) could also eliminate redundant sample preparations of 100% ^15^N-labeled and uniformly 100% [^13^C, ^15^N]-labeled samples of proteins. We determined the NMR structures of a small alpha-helical protein, the C domain of IgG-binding protein A from *Staphylococcus aureus* (SpaC), and a small beta-sheet protein, CBM64 module using [20% ^13^C, 100% ^15^N]-labeled sample and compared with the crystal structures and the NMR structures derived from the 100% [^13^C, ^15^N]-labeled sample. Our results suggest that one [20% ^13^C, 100% ^15^N]-labeled sample of small proteins could be routinely used as an alternative to conventional 100% [^13^C, ^15^N]-labeling for backbone resonance assignments, NMR structure determination, ^15^N-relaxation analysis, and ligand–protein interaction.

## 1. Introduction

NMR spectroscopy has been routinely used for elucidating three-dimensional structures of proteins in solution [1,2,3]. NMR structure determination can be advantageous over X-ray crystallography because it does not require any crystallization and can investigate protein structures under various solution conditions, including even in situ [4,5]. One critical bottleneck of NMR analysis compared with other three-dimensional analysis is the requirement of stable isotopic labeling such as ^15^N and ^13^C-labeling, which is typically desirable even for proteins as small as 5 kDa to speed up reliable NMR analysis. The isotopic labeling procedure for NMR inherently increases the cost and efforts for sample preparations, limiting the broader application of various NMR analysis. Even an NMR study of single-point mutants of a protein might require full NMR assignments of each variant due to the possible extensive changes in the NMR resonances. Stable isotope-labeled samples using ^15^N or/and ^13^C atoms are often prerequisites for such variants of a protein. Because of isotopic labeling, NMR analysis of several variants can quickly become time-consuming and costly for various useful NMR analyses, such as investigating protein–ligand interactions or protein dynamics. Even when their three-dimensional structures are already available, e.g., by crystallography or NMR, *de novo* NMR assignments can be cumbersome. Whereas NMR structure determination requires isotope-labeled samples and NMR assignments for all variants and their homologs of a protein, protein crystallography is more effective for elucidating such variants and homologs. Structural analysis of variants of a protein by crystallography is more commonly used because the molecular replacement method for phasing works more efficiently in such cases than NMR analysis when diffracting crystals can be obtained [6]. Therefore, protein crystallography is often used for structural analysis as the initial strategy, instead of NMR analysis. 

To take the best advantage of NMR spectroscopy, it is of practical importance to reduce effort, time, and costs to obtain NMR assignments and/or NMR structures of proteins with known structures [7,8]. Particularly for small proteins, NMR analysis could be faster and more practical than protein crystallography because of their smaller surface area available for crystal contacts in smaller proteins. Moreover, cryogenically cooled probes have significantly increased NMR sensitivity by a factor of 2–4 depending on solution conditions, requiring less protein material for the same measurement time [9]. In practice, a 0.5–1 mM protein solution, which has been used for small proteins using conventional probes, can now provide more than sufficient signal-to-noise (S/N) for small well-behaving proteins. As the sensitivities of NMR instruments such as ^13^C detection and cryoprobe keep improving further, it should be possible to lower ^13^C-isotope enrichment without any significant loss of the structural information. [U-^13^C_6_] d-glucose, which is typically used for the bacterial production of uniformly 100% [^13^C, ^15^N]-labeled proteins, accounts for approximately 80% of the chemical costs [10]. Reducing the amount of ^13^C-labeled glucose by a factor of four to five could reduce the chemical cost by a similar factor and investigate more proteins with the same price of a 100% [^13^C, ^15^N]-labeled sample. Fractionally ^13^C- and uniformly 100% ^15^N-labeled samples have been previously demonstrated for NMR backbone resonance assignments [10,11].

Here, we report NMR structure determination of the 86-residue cellulose-binding X-module from the *Spirochaeta thermophila* glycoside hydrolase (CBM64) and the 58-residue C-domain of protein A from *Staphylococcus aureus* (SpaC) using [20% 13C, 100% 15N]-labeling. We compared the NMR structures between the previously reported crystal structures and the NMR structures determined using a conventional 100% [13C, 15N]-labeled sample. We demonstrated that one sample using [20% ^13^C, 100% ^15^N]-labeling is sufficient for various NMR analysis of both small alpha-helical and beta-sheet proteins, including NMR structure determination. 

## 2. Results

### 2.1. NMR Assignments

NMR resonance assignment of the two proteins was carried out using a [20% ^13^C, 100% ^15^N]-labeled sample. For CBM64, we obtained 98.8% of the backbone amide ^15^N and ^1^H resonances, excluding the N-terminus and three proline residues (Figure 1A). For SpaC, 100% of the backbone amide resonances, excluding the N-terminus, were assigned (Figure 1B). Despite the fractional 20% ^13^C-labeling, we could assign 96.9% of the backbone atoms (H^N^, N^H^, H^α^, Cα, Cβ, and C′) and 93.6% of observable atoms of sidechains for CBM64 and 99.1% of the backbone atoms and 97.8% of side chains for SpaC. One of the disadvantages of the fractional 20% ^13^C-labeling scheme is the non-random breakage of ^13^C-^13^C bonds, deteriorating ^13^C-^13^C magnetization transfer in ^13^C-TOCSY pulse sequences such as in CC(CO)NH. Therefore, we had to rely on the HCCH-COSY or [^1^H, ^1^H]-TOCSY for the sidechain analysis in the case of CBM64. This analysis was supported by ^13^C-edited [^1^H, ^1^H]‑NOESY and ^15^N-resolved [^1^H, ^1^H]‑TOCSY. It was particularly problematic to assign ^13^Cγ atoms of leucine, glutamate, and glutamine by HCCH-COSY spectrum alone due to signal overlapping even for these small proteins. However, we still could connect the complete sequential connectivity in the same way as the conventional approach using a 100% [^13^C, ^15^N]-labeled sample. Decreased ^13^Cβ intensities in comparison with ^13^Cα were observed for most residues in CBCA(CO)NH, HNCACB, and intra-HNCACB experiments due to the amino-acid synthetic pathways (Appendix A, see below) [10,12]. However, these spectra still had enough signal-to-noise to perform sequential connections. It is still noteworthy that variations of ^13^Cβ intensities also contain the information about amino-acid types resulted from different biosynthetic pathways (Figure 2, Appendix A) [10]. We could also obtain stereospecific assignments from the fractional ^13^C-labeled sample for all methyl groups in Val and Leu residues based on ct-[^1^H, ^13^C]-HSQC experiment [13,14] (Figure 2A,B and Appendix A). Conventional 100% [^13^C, ^15^N]-labeling does not contain any information from the biosynthetic pathways of amino-acids as biosynthetically directed fractional ^13^C-labeling. Despite the reduced ^13^C fractions, the high completeness of the NMR assignments (>90%) suggests that it should be feasible to determine comparable NMR structures using a fractional ^13^C-labeled sample without conventional 100% ^13^C. ^15^N-labeling.

### 2.2. Effects of Fractional ^13^C-Labeling on NMR Spectra from the Different Biosynthetic Pathways

Biosynthetically directed fractional ^13^C-labeling using ^13^C_6_-d-glucose provides additional information about amino-acid types because every amino-acid has its specific biosynthetic pathway, giving rise to unique ^13^C-fine structures [10,12]. Reliable stereospecific assignment of diastereotopic methyl groups in leucine (Leu) and valine (Val) have been often achieved by fractional ^13^C-labeling, although it requires an additional sample (Figure 2A,B) [13,14]. 

Furthermore, it has been shown that fractional ^13^C-labeling facilitates assignments of aromatic NMR signals in Phe and Tyr because erythrose-4-phosphate and pyruvate pathway are involved in the biosynthesis of rings in Phe and Tyr (Figure 2C) [15]. Previously, we demonstrated that ^13^C-fine structures of carbonyl carbon ^13^C signals could be used to classify amino-acid types for resonance assignments [10]. 

One of the disadvantages of fractional ^13^C-labeling is that Cα-Cβ bond connections are no longer the intact bonds that originated from ^13^C_6_-glucose but depending on the amino-acid types. If Cα-Cβ bonds in all molecules were intact, NMR experiments using ^1^J_CαCβ_ coupling for the magnetization transfer between ^13^Cα and ^13^Cβ atoms, such as HNCACB, will basically give the same spectra as obtained from uniformly [^13^C, ^15^N]-labeled samples but with the reduced sensitivity of, e.g., 20%. If Cα-Cβ bonds from ^13^C_6_-glucose in all molecules were randomly mixed with unlabeled ^12^C_6_-glucose, the chance to have the ^13^Cα-^13^Cβ connectivity would be as low as 4% (for 20% fractional ^13^C-labeling). However, actual ^13^Cα-^13^Cβ bond connectivity is dependent on the biosynthetic pathway specific to amino-acid types [12]. 

To systematically investigate the effect of fractional ^13^C-labeling on ^13^Cα-^13^Cβ connectivity, we decided to take a simple approach for analyzing the ratios between peaks for ^13^Cα and ^13^Cβ atoms in a spectrum such as HNCACB. This is because the intensities of Cβ peaks from the fractional ^13^C-labeled samples are influenced by transverse nuclear relaxation, presences of passive couplings, magnetization transfer delay, fractions of ^13^C labeling, and aerobic conditions for protein expression. These factors modulating the peak intensities of Cβ correlation peaks make it challenging to postulate the outcomes for all proteins precisely. For example, intra-HNCACB does not eliminate sequential peaks because not all ^1^J_NC’_, ^2^J_NCα_, and ^1^J_NCα_ couplings are simultaneously active in all molecules like uniformly ^13^C,^15^N-labeled sample due to the fractional ^13^C-labeling [10]. Because inter-residual ^2^J_NCα_ coupling is conformation-dependent and smaller than ^1^J_NCα_, we summarized only the intensity ratios of intra-residual peaks (CB*_i_*/CA*_i_*) by using intra-residual Cα peaks as an internal reference out of the four expected peaks of sequential and intra-residual Cα, Cβ peaks (CA_*i*−1_, CA*_i_*, CB_*i*−1_, and CB*_i_*) (Figure 2D,E). 

In the CB*_i_*/CA*_i_* analysis, Phe, Tyr, and Ala residues constitute one group with the highest CB/CA ratio indicating that ^13^Cα-^13^Cβ connections are mostly intact from glucose via pyruvate (Figure 2C,F) [12]. Histidine residues, which are synthesized via the pentose phosphate pathway, also preserve the ^13^Cα-^13^Cβ connection judging from the high CB/CA ratio. In contrast to the high ratio group (Phe, Tyr, Ala, and His), Leu, Val, and Ile have the lowest ratio indicating that intact ^13^Cα-^13^Cβ bonds are broken during the biosynthesis (Figure 2A,B,F). Overall, Cβ peaks are only 10–25% weaker than Cα peaks for Phe, Tyr, His, and Ala (Figure 2E). Cβ peaks from Ile, Leu, and Val are 90% weaker than Cα peaks, setting the required S/N for the NMR experiments. 

Other amino-acid types are between the two abovementioned groups, depending on the amino-acid types. These residue-types except for Cys, Trp, and Ser are derived from oxalacetate or 2-oxoglutarate from Tricarboxylic Acid Cycle in *E. coli*, which can be easily affected by aerobic condition during protein expression. We did not have any data points for Met but expect to be similar to Asp, Asn, and Thr because Asp, Asn, Thr, and Met are biosynthesized from oxalacetate, which can be affected by the aerobic condition during protein expression. Thus, intensities of Cβ peaks might see more variations between different preparations for those amino-acid types unless the aeration condition is precisely controlled during protein production. Despite 10–90% signal reduction for the Cβ atoms, the weaker Cβ peaks or specific amino-acid types could offer additional information to classify amino-acid types constructively. 

### 2.3. NMR Structure Determination of CBM64 

We used an 86-residue carbohydrate-binding domain of the *Spirochaeta thermophila* glycoside hydrolase (CBM64) as a model β-sheet-rich small protein because the crystal structure is also available [17,18]. We obtained > 16 distance constraints per residue using 0.7 mM solution of [20% ^13^C, 100% ^15^N]-labeled CBM64, which seems to be enough for the high-resolution NMR structure (Table 1) [19]. The calculated NMR structures of CBM64 show a β-sandwich fold that is common among carbohydrate-binding proteins and is composed of nine β-strands (β1–β9) and a short 3_10_-helix-like turn (Figure 3A). Eight β-strands are involved in forming two distinct β-sheet surfaces (β3856-sheet with β3, β8, β5, and β6-strands and β2947-sheet with β2, β9, β4, and β7-strands). These two β-sheets are facing each other in an antiparallel manner. This arrangement results in the β-sandwich appearance. The short β1-strand is located at the N-terminus. β3856-sheet plays a vital role in the carbohydrate recognition with its hydrophobic surface, whereas the tyrosine-rich β2947-sheet shows higher electronegativity and is more vital for holding the structure fold [18,19]. Notably, both NMR structures of CBM64 obtained from uniformly labeled and fractionally labeled samples are nearly identical (Figure 3B), indicating that the fractionally ^13^C-labeled sample could produce sufficiently redundant distance constraints comparable to the uniform ^13^C-labeled samples within the accuracy of the NMR structure determinations (Table 2).

### 2.4. Comparison of the Structures Determined by Differently ^13^C-Labeled Samples of CBM64

For the NMR structure determination, we could use over 16 distance constraints per residue for CBM64 from both [20% ^13^C, 100% ^15^N]- and [100% ^13^C, ^15^N]-labeled samples, which seems more than sufficient to obtain reliable NMR structures [17]. Not surprisingly, we obtained about 10% fewer distance constraints from the 20% ^13^C-labeled sample than the 100% ^13^C-labeled sample, presumably due to the reduced S/N than 100% [^13^C, ^15^N]-labeled sample (Table 1). We compared the mean NMR structure obtained from [20% ^13^C, 100% ^5^N]-labeled sample (<
$\bar{20\%}$
>) with the mean structure obtained from 100% [^13^C, ^15^N]-labeled sample (<
$\bar{100\%}$
>) to assess the differences caused by dilution of ^13^C-labeling. For this analysis, we used the same chemical shift assignments but different NOE peak lists. Although the number of NOE distance restraints obtained for 100% [^13^C, ^15^N]-labeled sample is about 10% more, the structural statistics between the two NMR structures are very similar (Table 2). The RMSD values for the backbone atoms between <20%> and <100%> are slightly below 1.0 Å for residues 459–541 (Table 2). The RMSD values are similar throughout the backbone, although small variations can be observed for the unstructured region between residues 480–495 (Figure 3D). The accuracy of the distance constraints modulated by the degree of ^13^C-labeling at different sites, usually provided by upper distance constraints for the NMR structure determination, does not significantly influence the final structures by non-random fractional ^13^C-labeling.

### 2.5. Comparison of the NMR Structures with the Crystal Structures

We also compared the NMR structure of CBM64 with the previously reported crystal structures of CBM64 as the reference (Figure 3C). The five coordinates solved by X-ray crystallography without any ligand (PDB ID: 5E9P and 5E9O) were used for the comparison (Figure 3) [18]. The RMSD among the five coordinates (residues 459–541) is 0.23 ± 0.05 Å for the backbone atoms and 0.53 ± 0.21 Å for all the heavy atoms (<
$\bar{\mathrm{Xray}}$
>) (Figure 3B; Table 2) [18]. Both the crystal and NMR structures of CBM64 revealed almost identical three-dimensional structures with a β-sandwich fold (Figure 3). The RMSD between the mean structure of the NMR structure obtained from [20% ^13^C, 100% ^15^N]-labeled sample (<
$\bar{20\%}$
>) and the crystal structures (<Xray>) is 0.94 ± 0.05 Å as well as 1.23 ± 0.07 Å for the NMR structure from the 100% [^13^C, ^15^N]-labeled sample (<
$\bar{100\%}$
>) (Table 2). The better RMSD value for the fractionally ^13^C-labeled sample might be because the chemical shift assignments were used from the [20% ^13^C, 100% ^15^N]-labeled sample. The larger deviations between the crystal structure and two NMR structures are mostly originating from loop regions connecting well-defined β strands, such as a loop between β3 and β4 strands (Figure 3D). The structural deviations could be attributed to the difference between the crystal and solution structures.

### 2.6. NMR Structures of a Helical Protein, SpaC, and Comparison with the Crystal Structures

As CBM64 is a β-sheet protein containing only β-strands, we decided to test NMR structure determination by the same fractional ^13^C-labeling with another small alpha-helical protein, the C domain of *Staphylococcus aureus* protein A (SpaC), which is an IgG-binding protein [21]. We obtained more than 20 distance constraints per residue for the 58-residue SpaC despite the [20% ^13^C, 100% ^15^N]-labeled sample, which is more than the distance constraints obtained for CBM64 using a 100% [^13^C, ^15^N]-labeled sample despite the smaller size (Table 1). The higher number of distance constraints for SpaC can be attributed to the high concentration (5.6 mM) of the [20% ^13^C, 100% ^15^N]-labeled sample of SpaC. This result suggests that the degree of ^13^C labeling only influences the detection of NOEs peaks in the NOESY spectra, which can be satisfactorily compensated by increasing the sample concentration. The NMR structure of SpaC determined by [20% ^13^C, 100% ^15^N]-labeled sample shows a three-helix bundle fold with the RMSD of 0.3 Å (Figure 4A). The SpaC structure contains three α-helices (α1–α3) and one short 3_10_-helix-like turn as other domains of IgG binding protein A (Figure 4) [21]. We compared the solution NMR structure of SpaC determined by 20% ^13^C labeling with the two crystal structure coordinates of SpaC (PDB ID: 4NPD and 4NPE), indicating the RMSD of 1.3 Å (Figure 4B; Table 3) [21]. The differences are mainly located within loops connecting helices and near the N- and C-termini and could be partly caused by the flexibility of these regions in solution [22].

### 2.7. Interaction Analysis Using Fractional 20% ^13^C-Labeled Sample by NMR

Despite the requirement of isotope-labeling NMR, one advantage of NMR over X-ray crystallography is the possibility to study protein–ligand interactions under various solution conditions. CBM64 is one of more than 80 carbohydrate-binding modules (CBM) found in various cellulose-degrading enzymes from fungal and bacterial organisms [23]. CBM64 from *S. thermophila* binds to crystalline cellulose and also shows high thermostability and salt-tolerance [19]. Not surprisingly, the three-dimensional structure of CBM64 was previously reported by X-ray crystallography, making it less attractive to perform the structural analysis by NMR. There are often some three-dimensional structures with high-sequence identity for many small well-behaving proteins, owing to various structural genomics projects [24,25]. However, it would still be essential to investigate how different CBM modules interact with different carbohydrates under solution conditions even when three-dimensional coordinates are available. Whereas it will be challenging to crystallize CBM64 in the complex with crystalline cellulose that is not water-soluble, chemical shift perturbation (CSP) analysis using NMR titration experiments could identify interacting sites of celluloses when NMR assignments are readily available. Instead of crystalline cellulose, a short fragment of cellulose, D-cellobiose, is water-soluble and a suitable fragment to investigate possible binding sites of cellulose to CBM64.

We performed titration with D-cellobiose by recording [^1^H, ^15^N]-HSQC spectra of the [20% ^13^C, 100% ^15^N]-labeled CBM64 using D-cellobiose to observe chemical shift perturbations. Most of the amide resonances exhibited no chemical shift changes upon the addition of D-cellobiose even at 25:1 molar excess (Appendix A). Tiny but notable chemical shift changes (∆δ_av_ > 0.01 ppm) were observed only for two regions of CBM64, are the indole εNH side-chains corresponding to the four tryptophan residues in the β2947-sheet (W488, W495, W511, and W535) and the backbone amide signals for residues W488, S489, R490 and Y491 (Appendix A). This observation supports the previous report that the four tryptophan residues create the hydrophobic interface responsible for carbohydrate-binding via a coplanar linear arrangement [18,19]. Residues W488, S489, R490, and Y491 are located in the loop region connecting β3 and β4 strands. The largest chemical shift perturbation was 0.07 ppm for residue R490, even for the ^1^H dimension. Thus, we concluded that the small CSP values for residues 488–491 are likely to be due to the conformational changes upon the interaction with D-cellobiose or changes in dynamics of the four tryptophan residues rather than direct binding to residues 488-491 because these residues are located partially in the β3856-sheet and close vicinity of the carbohydrate-binding surface (Appendix A). The small CSP observed in the presence of D-cellobiose is in line with the weak interaction reported in the literature between type-A CBMs, including CBM64 and oligosaccharides, which is probably difficult to analyze by X-ray crystallography [26].

### 2.8. NMR Relaxation Analysis of the CBM64

Another advantage of NMR is the capability of probing dynamics such as internal motilities of proteins by ^15^N relaxation analysis once the NMR assignments are available. Therefore, we measured T_1_, T_2_, and ^15^N{^1^H}-NOE data using the [20% ^13^C, 100% ^15^N]-labeled sample (Appendix A). The average backbone T_1_ and T_2_ relaxation times are 694 ± 17 msec and 104 ± 7 msec for T_1_ and T_2_ without D-cellobiose, respectively. The T_1_/T_2_ values for 77 residues were 6.7 ± 0.7, which can be translated to the rotational correlation time, τ_c_ of 5.3 ± 0.3 nsec using the software DASHA [27]. The estimated τ_c_ is in good agreement with the empirical τ_c_, 5.4 nsec calculated from the molecular weight of 10018 daltons of CBM64 obtained from [28,29]. In the presence of 25 times excess of D-cellobiose, T_1_ and T_2_ are 636 ± 20 msec for and 113 ± 5 msec respectively, corresponding to τ_c_ of 5.3 ± 0.3 nsec. We observed overall small shifts of T_1_ and T_2_ in the presence of d-cellobiose, which was probably due to the change of the viscosity of the solution upon addition of 25 times excess of d-cellobiose (Appendix A). We detected notable differences in T_1_ and T_2_ in the presence of d-cellobiose around the loop regions connecting β3 and β4-strands as well as β7 and β8-strands, including a 3_10_-helix, where the largest differences are observed between the crystal and NMR structures. The lower values of ^15^N{^1^H}-NOE confirm the flexible N-terminal end observed in the NMR structures (Appendix A). T_1_/T_2_ values might indicate the regions where internal motions might have changed in the presence of d-cellobiose.

We found that notable differences in T_1_/T_2_ values in the presence of d-cellobiose are located around 3_10_-helix and in the loop connecting β3 and β4-strands (Appendix A). These regions coincide with the region identified by the CSP analysis (Appendix A). This observation suggests that the presence of D-cellobiose might have caused some changes in the internal motion or affect the relaxation rates due to the chemical exchanges. The 3_10_-helix is also located in the vicinity of residues 488–491, where detectable CSPs were observed for the amide groups. This coincidence of the changes might suggest that d-cellobiose caused the changes in the relaxation rates despite tiny chemical shift changes observed in the presence of a short fragment of cellulose.

## 3. Discussion

Technological advances in NMR instruments, such as cryogenically cooled probe heads and higher magnetic fields, have steadily increased the sensitivity of NMR spectrometers [9,30]. Such sensitivity improvement has lowered sample requirement at a fixed measurement time to reach the same S/N ratio. However, labeled sample preparation and downstream resonance assignments are still the major bottlenecks for NMR studies of proteins, including protein–ligand interactions and protein dynamics. Even for small well-behaving proteins, sample preparation and backbone resonance assignment could be laborious yet requiring protein NMR expertise. NMR studies of variants with point mutations, homologous proteins, and proteins with known structures can still be as time-consuming as de novo NMR analysis of a protein.

Here we demonstrated that one fractional 20% ^13^C- and 100% ^15^N-labeled sample for small proteins is sufficient for most of the NMR analysis, including backbone NMR assignment and NMR structure determination. The NMR structures of a 7-kDa α-helical and a 10-kDa β-sheet proteins were determined with high accuracy using fractional 20% ^13^C- and 100% ^15^N-labeling without significant loss of the structural information in comparison to conventional uniform 100% [^13^C,^15^N]-labeling and the crystal structures. Once NMR assignments are available, NMR could provide beneficial information such as protein–ligand interactions and protein dynamics very efficiently under various solution conditions.

Dilution of [U-^13^C_6_]-d-glucose by a factor of five could lower the chemical cost by a similar factor [10,11]. The isotope cost for one [20% ^13^C, 100% ^15^N]-labeled sample would be about one-fourth of the cost for one 100% [^13^C, ^15^N]-labeled sample, assuming the following amounts and prices for 1.0 L of minimal M9 medium: 0.8 g for ^15^NH_4_Cl (25 EUR/g) and 0.4 g [U-^13^C_6_]-d-glucose (100 EUR/g). Moreover, with only twice the cost required for one 100% ^15^N-labeled sample, one could obtain the same structural information as a 100% [^13^C, ^15^N]-labeled sample by adding 20% [U-^13^C_6_]-d-glucose to 100% ^15^N-labeled sample. Fractionally [20% ^13^C, 100% ^15^N]-labeled samples can serve as 100% ^15^N-labeled samples because only less than 10% of side-bands from ^13^C atoms could appear in the [^1^H,^15^N]-HSQC spectrum even if ^13^C resonances were not decoupled [10]. Moreover, the same 20%^13^C-labeled sample provides 20 times better ^13^C sensitivity when ^13^C detection is used. Despite the five-fold dilution of ^13^C atoms, we could successfully obtain not only NMR assignments but also the three-dimensional structures of a 10-kDa protein (CBM64) using ca. 2 mg of one [20% ^13^C, 100% ^15^N]-labeled sample [10,11].

We demonstrated that the NMR structures determined by the fractionally ^13^C-labeling scheme are comparable to the previously determined crystal structures. Moreover, the differences between the NMR structures obtained from the fractional ^13^C-labeling and the conventional uniform ^13^C-labeling are marginal, suggesting the redundant upper-distant constraints derived from NOEs when S/N is sufficient. In addition to the lower cost, fractionally ^13^C-labeled samples could provide some additional advantages when signal-to-noise is sufficient for NOE analysis and backbone resonances assignments. Non-random breakage of ^13^C-^13^C bonds due to different metabolic pathways of 20 amino-acid types could result in different ^13^C-^13^C patterns specific to each amino-acid type, which can be exploited for stereospecific assignments of diastereotopic methyl groups as well as for the classifications of amino-acid types in triple resonance spectra such as HNCO and HNCACB [10,13]. Lack of passive ^13^Cα-^13^Cβ scalar coupling might also contribute to improved resolution and sensitivity [10,31]. These advantages are usually lost when 100% [^13^C, ^15^N]-labeled samples are used. In our laboratory, we routinely produce [20% ^13^C, 100% ^15^N]-labeled samples instead of 100% ^15^N-labeled samples because it has almost full capabilities to perform the majority of NMR experiments at twice the chemical cost of 100% ^15^N-labeling, as demonstrated in this article. This labeling strategy could also avoid multiple redundant preparations of labeled samples for small well-behaving proteins. We believe that the fractional 20% ^13^C- and 100% ^15^N-labeling scheme could benefit from more sophisticated labeling methods such as segmental isotopic labeling to study a small domain in the full-length context [32].

## 4. Materials and Methods

### 4.1. Cloning, Protein Production, and Purification for NMR Studies

CBM64 (residues 456–541) of *S. thermophila* cellulase GH5 was expressed in *E. coli* strain ER2566 cells (New England Biolabs) transformed with the plasmid pBHRSF274 encoding the N-terminally His-tagged SUMO-CBM64 fusion. The [20% ^13^C, 100% ^15^N]-labeled sample was prepared by expressing the fusion protein in 2 L M9 medium supplemented with 25 μg/mL kanamycin, [U-^13^C_6_] d-glucose (0.42 g/L) (Cambridge Isotope Laboratories, Inc.) and natural isotope abundance d-glucose (1.6 g/L) as sole carbon source, and ^15^NH_4_Cl (0.8 g/L) as a sole nitrogen source. The 100% [^13^C, ^15^N]-labeled sample was prepared by expressing the protein in 2 L M9 medium containing [U-^13^C_6_]-d-glucose (2.0 g/L) and ^15^NH_4_Cl (0.8 g/L). The expressed fusion proteins were purified by ion metal chromatography (IMAC) followed by removing His-tagged SMT3 as previously described [33]. The purified protein was dialyzed against 20 mM sodium phosphate buffer (pH 6.0) and concentrated to a final volume of 250 μL (0.73 mM [20% ^13^C, 100% ^15^N]-labeled sample and 0.6 mM 100% [^13^C, ^15^N]-labeled sample). The protein samples containing 5% D_2_O (*v*/*v*) were transferred into 5.0 mm microcell NMR tubes (Shigemi Inc.) with comparable molar amounts (185 nmol (2.0 mg) for [20% ^13^C, 100% ^15^N]-labeled sample and 150 nmol (1.6 mg) for 100% [^13^C, ^15^N]-labeled sample).

The gene of C domain of Protein A (SpaC), which is a 42 kDa surface protein found in the cell wall of the *Staphylococcus aureus*, was chemically synthesized and purchased from Integrated DNA Technologies Inc (Iowa, USA) and cloned into pHYRSF53 using two restriction sites of *Bam*HI and *Kpn*I, which resulted in plasmid pBHRSF212 for the protein expression of a His-tagged SUMO fusion [33]. [20% ^13^C, 100% ^15^N]-labeled SpaC was expressed and purified as above and concentrated into 5.6 mM in 20 mM sodium phosphate buffer (pH 6.0) buffer for NMR measurements.

### 4.2. Multidimensional NMR Spectroscopy

The following 2D and 3D experiments were used: [^1^H, ^15^N]-HSQC, BEST-intra-HNCO, BEST-intra-HNCA, BEST-intra-HNCACB, and CBCA(CO)NH [34,35]. ^1^H and ^13^C assignments for aliphatic side-chain were based on [^1^H, ^13^C]-HSQC, HCCH-COSY, ct-[^1^H, ^13^C]-HSQC, ^13^C-edited [^1^H, ^1^H]‑NOESY, and ^15^N-edited [^1^H, ^1^H]-TOCSY, for backbone resonance assignments of CBM64,. The assignments of aromatic sidechains were based on [^1^H, ^13^C]-HSQC, ^15^N-edited [^1^H, ^1^H]-TOCSY, ^15^N-edited [^1^H, ^1^H]-NOESY, and ^13^C-edited [^1^H, ^1^H]-NOESY. For SpaC backbone assignments, the following spectra were used: [^1^H, ^15^N]-HSQC, CON, CACO, HNCA, intra-HNCA, HNCACB, HNCO, intra-HNCO. We also recorded ^13^C-detected spectra, benefiting from the high concentration of the SpaC sample [36]. Aliphatic sidechain assignment of ^1^H and ^13^C resonances was performed using [^1^H, ^13^C]-HSQC, HCCH-COSY, CACO, ^13^C-edited [^1^H, ^1^H]-NOESY, and ^15^N-edited [^1^H, ^1^H]-NOESY. [^1^H, ^13^C]-HSQC and ^13^C-edited [^1^H, ^1^H]-NOESY were used for aromatic sidechain assignments. The spectra were analyzed using CcpNmr Analysis 2.4.2 software [37].

### 4.3. NMR Data Acquisition and Processing

All NMR experiments for both [20% ^13^C, 100% ^15^N]-labeled and 100% [^13^C, ^15^N]-labeled samples of CBM64 were recorded at 303 K on a Bruker 850 MHz Avance III HD NMR spectrometer equipped with a cryogenically cooled TCI probe head at the ^1^H frequency of 850 MHz. For the BEST-intra-HNCA experiment, a total of 160 (ω_1_) × 64 (ω_2_) × 800 (ω_3_) complex points were collected with t_1, max_ = 11.7 msec, t_2, max_ = 14.2 msec, and t_3, max_ = 36.3 msec, respectively. The total experiment time was 11 h. For the BEST-intra-HNCACB experiment, a total of 140 (ω_1_) × 64 (ω_2_) × 800 (ω_3_) complex points were collected yielding t_1, max_ = 4.1 msec, t_2, max_ = 10.9 msec, and t_3, max_ = 36.3 msec, respectively. The total experiment time was 11 h. For the BEST-intra-HNCO experiment, a total of 128 (ω_1_) × 64 (ω_2_) × 800 (ω_3_) complex points were collected with t_1, max_ = 21.4 msec, t_2, max_ = 14.2 msec, and t_3, max_ = 36.3 msec, respectively. The total experiment time was 10 h. For the CBCA(CO)NH experiment, a total of 140 (ω_1_) × 64 (ω_2_) × 1960 (ω_3_) complex points were collected with t_1, max_= 4.1 msec, t_2, max_= 10.9 msec, and t_3, max_= 88.8 msec, respectively. The total experiment time was 22 h. For the ^15^N dimension (ω_1_), a linear prediction of 16 forward complex points was applied. Shifted sine bell-window function (QSINE with SSB = 2) was applied, and the data were zero-filled to give a final matrix of 512 (ω_1_) × 256 (ω_2_) × 2048 (ω_3_) complex points. NMR spectra were processed using Topspin 3.2. One [20% ^13^C, 100% ^15^N]-labeled sample of SpaC was used for all the NMR experiments.

### 4.4. NMR Structure Calculations

Three-dimensional NMR structures were calculated in the same way for both CBM64 and SpaC using CYANA 3.97 software, based on the automated NOE analysis algorithm [38,39,40,41]. Upper distance restraints were derived from the 3D ^15^N- and ^13^C-edited [^1^H, ^1^H]-NOESY spectra with 75-ms mixing time. Backbone torsion angle restraints from chemical shifts were generated using TALOS-N software. [42,43]. No additional hydrogen bond restraints were introduced. Energy minimization in explicit waters was performed for the 20 best CYANA conformers with the lowest CYANA target function using AMBER14 [44]. The structures were validated with PSVS 1.5. [16]. The structures were visualized with MOLMOL [20].

The structural coordinates and the chemical shifts of the [20% ^13^C, 100% ^15^N]-labeled and 100% [^13^C, ^15^N]-labeled samples of CBM64 and SpaC were deposited in the protein data bank [45] (PDB ID code: 6FFU, 6FFQ, and 6SOW) and BMRB [46] (accession number: 34229, 34227, and 34430).

### 4.5. ^15^N-Nuclear Relaxation Analysis of CBM64

For ^15^N relaxation analysis, the longitudinal (T_1_) and transverse (T_2_) relaxation times together with ^15^N{^1^H}-NOEs for backbone ^15^N atoms were recorded at 303 K using well-established pulse sequences [47,48]. T_1_ and T_2_ relaxation times were obtained using the following series of delays: 10, 50, 100, 200, 300, 500, 800, 1000, 1200, 2000 msec for T_1_, and 16, 64, 96, 128, 156, 196, 224, 256 msec for T_2_. Recycle delay of 3.0 sec was used for T_1_ experiments and of 2.0 sec for T_2_ relaxation experiments. T_1_ and T_2_ relaxation times were estimated by fitting a single exponential decay to the signal intensities: *I(t)* = *I*_0_ × exp(−*t/T*_1, 2_), where I(*t*) = is the signal intensity after a delay of time *t* and *I*_0_ = signal intensity at *t* = 0. T_1_ and T_2_ relaxation data were processed and analyzed using Topspin Dynamic Center software (version 2.1.8, Bruker, USA). ^15^N{^1^H}-NOE values were obtained from the signal intensity ratio (*η* = *I/I*_0_) acquired with and without proton saturation during recycling delay (5.1 sec), where *I* and *I*_0_ are the measured signal intensities in the presence and absence of proton saturation, respectively. The volumes of the signals of the spectra were analyzed and fitted using CcpNmr Analysis 2.4.2 software [37].

### 4.6. NMR Titrations of D-Cellobiose with CBM64

Titrations with d-cellobiose (Sigma-Aldrich, CAS 528-50-7, St. Louis, MO, USA) were conducted to investigate carbohydrate interaction with CBM64. The ligand was chosen instead of crystalline cellulose, as it is water-soluble and an easily accessible fragment of cellulose. Carbohydrate concentrations corresponding to molar ratios (ligand: protein) of 0:1, 0.25:1, 1:1, 5:1, 12.5:1, and 25:1 were used for chemical shift perturbation experiments. A [^1^H, ^15^N]-HSQC spectrum was recorded at each titration point. Average chemical shift perturbations (CSP) were calculated using the equation, (∆δ_av_ = [(δH^N^)^2^ + (0.154 × δN^H^)^2^]^1/2^) [49].

## 5. Conclusions

We demonstrated that one [20% ^13^C, 100% ^15^N]-labeled sample for small well-behaving proteins is sufficient to obtain backbone resonance assignments, NMR structures, and other analyses such as ^15^N relaxation data. The NMR structures determined by [20% ^13^C, 100% ^15^N]-labeled sample are in good agreement with the crystal structures and the NMR structures determined by the conventional 100% [^13^C, ^15^N]-labeled sample. We propose [20% ^13^C, 100% ^15^N]-labeling as an alternative to 100% [^13^C, ^15^N]-labeling for routine sample preparations of proteins for various NMR studies to avoid redundant sample preparation, particularly for small well-behaving proteins. This fractional ^13^C-labeling scheme would be advantageous for NMR studies of proteins such as variants by saving sample preparation steps and chemical cost, yet containing additional information from the biosynthetic pathway. This alternative approach will be more applicable when NMR sensitivities are further improved, such as ultra-higher magnetic fields and direct ^13^C or ^15^N detection.

## Figures and Tables

**Figure 1 molecules-26-00747-f001:**
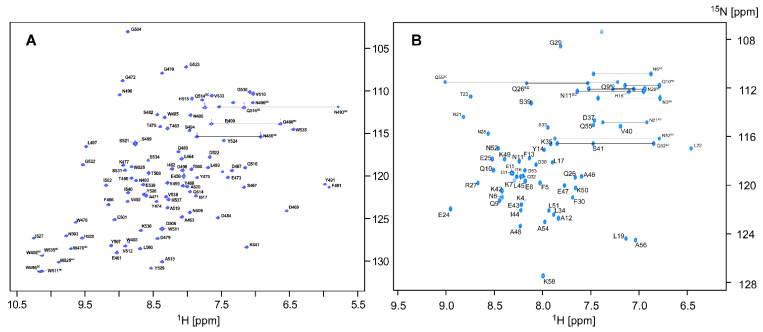
(**A**) Two-dimensional [^1^H, ^15^N]-HSQC spectrum of [20% ^13^C, 100% ^15^N]-labeled CBM64 recorded at the ^1^H frequency of 850 MHz, 303 K with the assignment using one letter amino-acid codes (residues 458–541). (**B**) Two-dimensional [^1^H, ^15^N]-HSQC spectrum of [20% ^13^C, 100% ^15^N]-labeled SpaC with the assignment using one letter amino-acid codes (residues 4–58). The sidechains are marked by “sc”. Identified sidechain amides of glutamine (Q^sc^) and asparagine (N^sc^) are indicated by horizontal lines.

**Figure 2 molecules-26-00747-f002:**
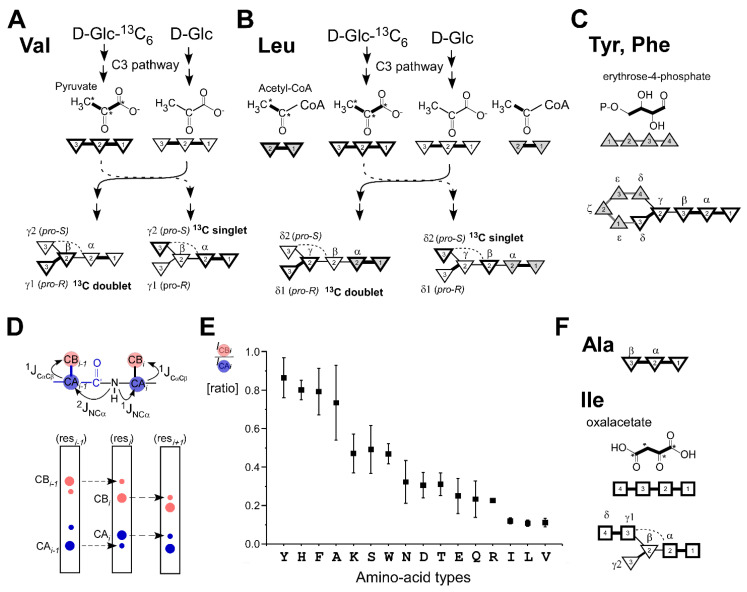
Effects of fractional ^13^C labeling on NMR spectra. Schematic presentation of reaction pathways for the biosynthesis of valine (Val) (**A**) and leucine (Leu) (**B**) from a mixture of U-^13^C_6_ glucose and unlabeled glucose, showing the stereochemistry [13]. Asterisks indicate ^13^C atoms. ^13^C-singlets and ^13^C-doublets are indicated for *pro-S* and *pro-R* methyl groups in Val and Leu [13]. (**C**) Backbone skeletons of Tyr and Phe. The intermediate of erythrose-4-phosphate is shown and also presented schematically as four shaded triangles as carbon atoms connected with lines. Thick lines indicate the intact bond connection from glucose molecules. (**D**) J couplings used for sequential backbone assignments and sequential walks using strips from 3D spectra such as HNCACB, CBCANNH, and others. (**E**) A plot of the ratios of the intensities between Cα and Cβ (CB*i*/CA*i*) from HNCACB (SpaC) and intra-HNCACB (CBM64) against amino-acid types. Glycine (Gly) and proline (Pro) and the data with overlapped signals are removed from the analysis. In total, we used the following number of the data points for amino-acid types: Y (9), H (3), F (4), A (10), K(8), S(6), W(5), N (9), D (9), T (6), E (10), Q (8), R (1), I (7), L (10), and V (6), showing the amino-acid type in one-character codes with the number of data points in parentheses. (**F**) Backbone skeletons of Ala and Ile. Ala is biosynthesized via the pyruvate pathway. Ile is biosynthesized from oxaloacetate from the tricarboxylic acid (TCA) cycle [12]. Dashed lines indicate the fragment arising from the same intermediate molecule [12]. Inverse triangles indicate the carbon skeleton of the C3 unit of pyruvate intermediate from glycolysis [12]. The carbon skeletons of acetyl-CoA intermediates used for the biosynthesis of Leu are also shown in shaded inverse triangles. Open rectangles are carbon skeletons of the intermediate of the TCA cycle. Shaded triangles indicate the carbon skeleton from the pentose phosphate pathway. Thick lines indicate ^13^C-labeled carbons or intact bonds from glucose. Greek alphabets show the positions in amino-acids.

**Figure 3 molecules-26-00747-f003:**
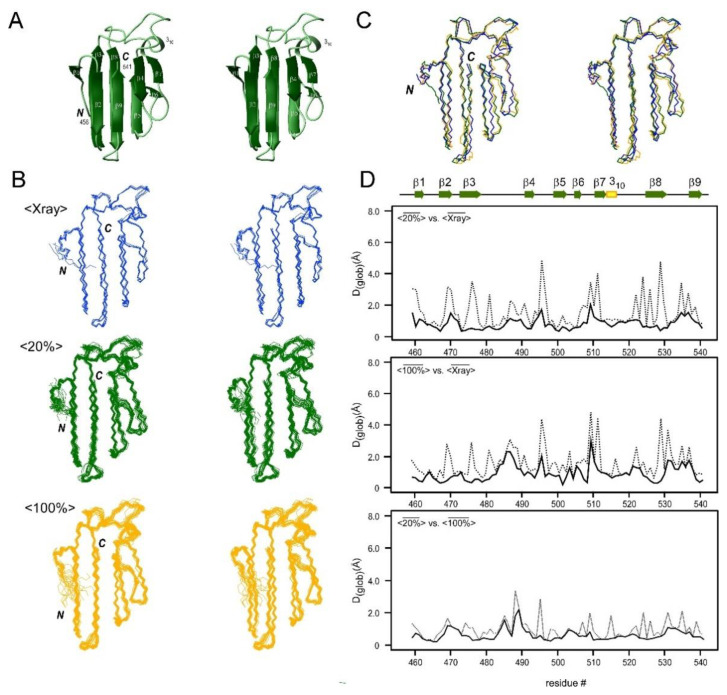
(**A**) A stereoview of the cartoon model of the lowest energy NMR conformer of CBM64 determined using the [20% ^13^C, 100% ^15^N]-labeled sample. (**B**) Stereoviews of bundles of the crystal structures (<Xray>, colored in blue), the 20 NMR conformers determined using the [20% ^13^C, 100% ^15^N]-labeled sample (<20%>, colored in green) and the 20 NMR conformers from 100% [^13^C, ^15^N]-labeled sample (<100%>, colored in orange) of CBM64. *N* and *C* indicate the C- and N-termini, respectively. (**C**) Stereoview of the superposition of the mean structure of crystal structures (<
$\bar{\mathrm{Xray}}$
>, blue), and the mean NMR structure from the [20% ^13^C, 100% ^15^N]-labeled sample (<
$\bar{20\%}$
>, green) and the mean NMR structure from the 100% [^13^C, ^15^N]-labeled sample (<
$\bar{100\%}$
>, orange) of CBM64. (**D**) Plots of the global displacement (*D*_(glob)_) (Å) among the three mean structures for CBM64 of <
$\bar{\mathrm{Xray}}$
>, <
$\bar{20\%}$
>, and <
$\bar{100\%}$
> for residues 459–541. Plots of the global displacement for <
$\bar{\mathrm{Xray}}$
> *versus* <
$\bar{20\%}$
>, <
$\bar{\mathrm{Xray}}$
> *versus* <
$\bar{100\%}$
>, and <
$\bar{20\%}$
> *versus* <
$\bar{100\%}$
> are shown on the top, middle, and bottom, respectively. Backbone heavy atoms (C^α^, C′, N) are indicated with a solid line and heavy atoms (all C, N, O) with a dashed line. The secondary elements are shown above the plots.

**Figure 4 molecules-26-00747-f004:**
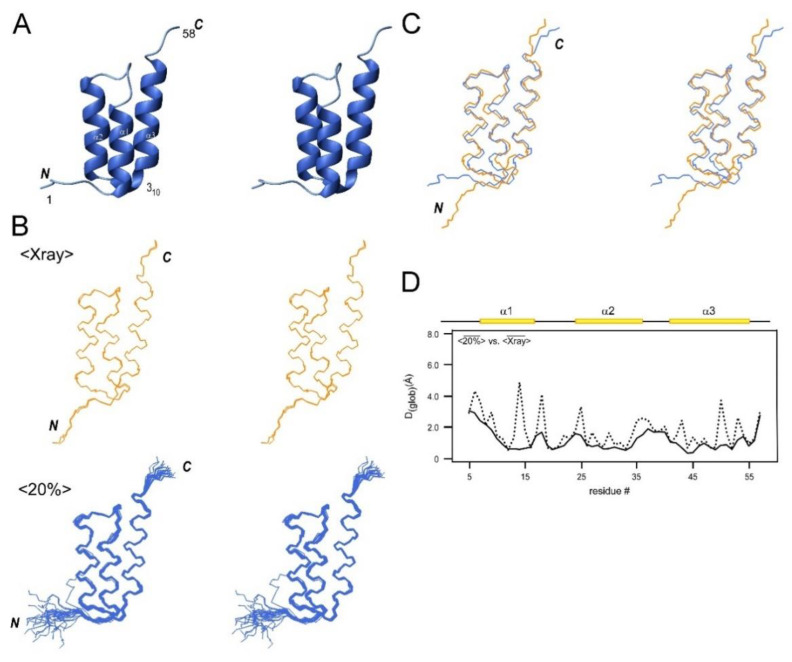
(**A**) Stereoview of the cartoon model of the lowest energy NMR conformer of SpaC determined using the [20% ^13^C, 100% ^15^N]-labeled sample. (**B**) Stereoview of bundles of the crystal structures (<Xray>, colored in orange) and the 20 NMR conformers determined using the [20% ^13^C, 100% ^15^N]-labeled sample (<20%>, colored in blue) of SpaC. The C- and N-termini are indicated. (**C**) Stereoview of the superposition of the mean structure of crystal structures (<
$\bar{\mathrm{Xray}}$
>, orange) and the mean NMR structure from the [20% ^13^C, 100% ^15^N]-labeled sample (<
$\bar{20\%}$
>, blue) of SpaC. (**D**) The global displacement (*D*_(glob)_) (Å) among the mean structures of <
$\bar{\mathrm{Xray}}$
>, and <
$\bar{20\%}$
> for residues 5–57. Backbone heavy atoms (C^α^, C′, N) are indicated with a solid line and heavy atoms (all C, N, O) with a dashed line. The secondary structures are shown above the plots.

**Table 1 molecules-26-00747-t001:** Structural Statistics for the 20 Energy-Minimized NMR Conformers of CBM64 and SpaC.

Protein	CBM64	CBM64	SpaC
Completeness of resonance assignments (%) ^a^	[20% ^13^C, 100% ^15^N]	100 % [^13^C, ^15^N]	[20% ^13^C, 100% ^15^N]
Backbone	96.9	96.9	99.1
Side-chain Aromatic	93.6 88.5	93.6 88.5	97.8 84.6
Stereospecific methyl	100	100	100
Distance restraints			
Total	1396	1575	1163
Sequential (|i − j| ≤ 1)	680	752	544
Medium range (1 < |i − j| < 5)	178	196	357
Long range (|i − j| ≥ 5)	538	627	262
No. of restraints per residue	16.2	18.3	20.1
No. of long-range restraints per residue	6.3	7.3	4.5
Residual restraint violations			
Average no. of distance violation per structure			
0.1–0.2 Å	5.5	6.7	8.3
>0.2 Å	0 (max 0.20)	0.2 (max 0.26)	0.3 (max 0.25)
No. of dihedral angle violations per structure >5°	0	0	0
Model quality ^b^			
Rmsd backbone atoms (Å)	0.4	0.4	0.3
Rmsd heavy atoms (Å)	0.8	0.9	0.7
Rmsd bond lengths (Å)	0.015	0.015	0.013
Rmsd bond angles (°)	2.2	2.1	2.0
MolProbity Ramachandran statistics ^b^			
Most favored regions (%)	95.6	95.4	98.5
Allowed regions (%)	4.3	2.3	1.5
Disallowed regions (%)	0.1	2.3	0
Global quality scores (raw/Z score) ^b^			
Verify3D	0.41/−0.80	0.42/−0.64	0.35/−1.77
ProsaII	0.39/−1.08	0.43/−0.91	1.23/2.40
PROCHECK (φ − ψ)	−0.51/−1.69	−0.49/−1.61	0.18/1.02
PROCHECK (all)	−0.59/−3.49	−0.58/−3.43	−0.00/−0.47
MolProbity clash score	1.20/1.32	2.95/1.02	1.51/1.27
Model contents			
Ordered residue ranges	459−541	459−541	5−57
Total no. of residues	86	86	58
BMRB accession number	34229	34227	34430
PDB ID code	6FFU	6FFQ	6SOW

^a^ Calculated from the expected number of resonances, excluding highly exchangeable protons (N-terminal, Lys, amino and Arg guanidino groups, hydroxyls of Ser, Thr, and Tyr), carboxyls of Asp and Glu, and unprotonated aromatic carbons. Backbone: H^N^, N^H^, Cα, Cβ, Hα, C′. ^b^ Calculated using PSVS version 1.5 [16].

**Table 2 molecules-26-00747-t002:** The Comparison of the Crystal and NMR Structures of CBM64.

	RMSD (Å) for Residues 459–541 of CBM64 ^a^		
Mean	<Xray>	<20%>	<100%>
$<\bar{\mathrm{Xray}}>$	0.23 ± 0.05 (0.53 ± 0.21)	0.97 ± 0.09 (2.12 ± 0.10)	1.11 ± 0.13 (2.04 ± 0.12)
$<\bar{20\%}>$	0.94 ± 0.05 (2.00 ± 0.07)	0.37 ± 0.09 (0.77 ± 0.11)	0.98 ± 0.15 (1.70 ± 0.12)
$<\bar{100\%}>$	1.23 ± 0.07 (1.97 ± 0.09)	0.93 ± 0.10 (1.65 ± 0.11)	0.42 ± 0.13 (0.81 ± 0.21)

^a^ RMSD (Å) values are shown for backbone atoms (C^α^, C′, N) and in brackets for heavy atoms (all C, N, O). The values were calculated using MOLMOL [20].

**Table 3 molecules-26-00747-t003:** The Comparison of the Crystal and NMR Structures of SpaC (RMSD (Å)).

	RMSD (Å) for Residues 5–57 of SpaC ^a^	
Mean	<Xray>	<20%>
$<\bar{\mathrm{Xray}}>$	0.09 ± 0.00 (0.30 ± 0.00)	1.34 ± 0.13 (2.23 ± 0.14)
$<\bar{20\%}>$	1.32 ± 0.02 (2.14 ± 0.04)	0.30 ± 0.06 (0.68 ± 0.10)

^a^ RMSD (Å) values are shown for the backbone atoms (C^α^, C′, N) and in brackets for heavy atoms (all C, N, O). The values were calculated using MOLMOL [20].

## Data Availability

Protein coordinates were deposited at protein data bank (PDB) with accession ID codes: 6FFU, 6FFQ, and 6SOW and Biological Magnetic Resonance Data Bank (BMRB) with accession numbers: 34229, 34227, and 34430.

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
