# Peer review of "NMR Structure Determinations of Small Proteins Using only One Fractionally 20% 13C- and Uniformly 100% 15N-Labeled Sample"

_molecules, 2021, doi:10.3390/molecules26030747_

Round 1

Reviewer 1 Report

This manuscript is well written although originality and scientific importance of this study is low.

The reviewer would like to recommend minor revision to improve quality of this manuscript, commented as below:

 (Lines No. 100-102) The author described that "We could also obtain dia-stereospecific assignments from the fractional 13C-labelled sample for all methyl groups in valine and leucine residues based on the ct-[1H,13C]-HSQC experiment [13]".  The authors have to show the ct-[1H,13C]-HSQC spectra of all samples as Supplemental Figures in order to demonstrate how fractional 13C-labelling and dia-stereospecific assignment were actually achieved.  The reviewer does not understand whether the fractional 13C-labelling and dia-stereospecific assignment of methyl groups of valine and leucine is possible by "20%-labeling of 13C".  If it is true, it is expected that scientific importance and utility of the methodology of this study will be significantly increased because this methodology become hopeful for NMR structure analyses for more big (> 150 aa) proteins.

Author Response

Responses to the reviewer #1:

>This manuscript is well written although originality and scientific importance of >this study is low.

Response: We appreciate this criticism and agree that our article does not report spectacular results for most of the readers, but a critical step to use this labeling scheme for routine NMR structure determination by assessing the effects of fractional labeling on the NMR structures, which someone needs to perform and report at one point.

>The reviewer would like to recommend minor revision to improve quality of this >manuscript, commented as below:

> (Lines No. 100-102) The author described that "We could also obtain dia->stereospecific assignments from the fractional 13C-labelled sample for all ?>methyl groups in valine and leucine >residues based on the ct-[1H,13C]-HSQC ?>experiment [13]".  The authors have to show the ct-[1H,13C]-HSQC spectra of ?>all samples as Supplemental Figures in order to demonstrate how fractional >13C-labelling and dia-stereospecific assignment were actually achieved.  The >reviewer >does not understand whether the fractional 13C-labelling and dia->stereospecific assignment of methyl groups of valine and leucine is possible by >"20%-labeling of 13C".  If it is true, it is >expected that scientific importance >and utility of the methodology of this study will be >significantly increased >because this methodology become hopeful for NMR structure analyses for >more >big (> 150 aa) proteins.

Response: Biosynthetically directed 13C fractional labeling is a robust and well-established method to stereo-specifically assign methyl groups in leucine and valine because the pro-R methyl group has 13C-doublet whereas the pro-S methyl group is 13C-singlet (Neri et al. 1989 and Szypersiki, 1995). In the ct-HSQC spectrum, the methyl carbon signals are modulated with the term Cosn1JccT), where T is the constant time evolution period, and n is the number of 13C atoms attached. When T is set to 1/Jcc (27msec), pro-S and pro-R methyl groups have a different sign in the ct-HSQC spectrum. The longer constant transfer time (T) used in ct-HSQC spectrum might not be suitable for larger proteins due to shorter 13C relaxation time of larger proteins.

We added a new section of 2.2 to explain the detailed effects on Cβ caused by the biosynthetic pathways, including methyl groups. We also added Figure S2B as a supplemental figure with the assignments for pro-S methyl groups as requested. 

Reviewer 2 Report

This manuscript is trying to address lowering the cost of production of C13 labeled protein for NMR experiments. Considering the major cost for protein NMR structural studies is the instrumental time cost, the significance of the work is limited. Nevertheless, this paper provides a method to lower the cost of C13 labeling, make protein NMR relatively more affordable.  I suggest to make a statement that this should be a second option as the method is only comparable to fully labeling at the best.

It is not clear how 20% labeling affects overall spectra, especially Cb on HNCACB. As this is crucial, I suggested to compare the spectra at same condition to show the change, or even better, semi-quantitatively. There are also two extra Ca peaks on 20% C13 labeling of the left panel S1, which is strange and needs explanation.

More detailed information would be appreciated about amino acid type based on variation of Cb peak intensity. For what type of amino acids? how different the intensity compares to the normal one?

Overall this is a well-written manuscript with high quality data supporting the conclusion. I recommend accept with minor revision. 

Author Response

Responses to the reviewer #2:

>This manuscript is trying to address lowering the cost of production of C13 >labeled protein for >NMR experiments. Considering the major cost for protein >NMR structural studies is the >instrumental time cost, the significance of the >work is limited. Nevertheless, this paper provides a method to lower the cost of >C13 labeling, make protein NMR relatively more affordable.  I suggest to >make a statement that this should be a second option as the method is only >comparable to fully labeling at the best.

We added the suggested statement in the text.  

However, we mostly disagree with this opinion as it depends on the perspectives, comparing two completely different costs that can not be compared directly and are depending on the environment. 

Without a sample, an NMR spectrometer would be idle with the same running cost and depreciation. Occasional samples to keep the NMR spectrometer busy would make the instrumental time more expensive because maximum instrument time is not fully utilized. If there were always long queues for protein NMR with many more important samples, more NMR spectrometers might be sold (or funded) and become cheaper in the long run.

   If one must spend 1000 EUR to prepare one sample, this could increase the hurdle for a student to prepare a sample because sample preparation does not always succeed. Students and postdocs would try to prepare more labeled samples when the hurdle is lowered to 250 or 100EUR, particularly when it gives the same structural information. We are also suggesting one labeled sample preparation, instead of two 15N-labeled and doubly 13C, 15N-labeled (DL) samples from E.coli (I do not know any group where people prepare only DL-samples from the beginning except for cell-free production),  thereby saving the working hours by half, which can be much more expensive than the instrumental time but often ignored. 

Therefore, the total estimated "cost" strongly depends on the viewpoints and to whom one asks.

>It is not clear how 20% labeling affects overall spectra, especially Cb on >HNCACB. As this is crucial, I suggested to compare the spectra at the same >condition to show the change, or even better, >semi-quantitatively. There are >also two extra Ca peaks on 20% C13 labeling of the left panel S1, which is >strange and needs explanation.

Response: This is a good and important point, which we did not mentioned in the previous manuscript. Overall, fractional 20% 13C-labeling does not affect very much because of the low digital resolution typically used for 3D spectrum hiding fine 13C structures, but reduced S/N by one fifth. As pointed out, non-random distributions of 13C atoms by biosynthesis of amino-acids give the certain patterns specific to AA-types (see new section 2.2). As for the extra peaks, intra-HNCACB spectrum was shown instead of HNCACB, which is now corrected. In the intra-HNCACB experiment, all  1JNC’, 1JNCα, 2JNCα are assumed to be present in all molecules for eliminating intra-residual signals. However, with the fractional 13C-labeling, this is no longer valid but depending on the molecules because 13C-fractional labeling creates a mixture of isotopomers with different 13C-labeled patterns. Therefore, sequential signals appear unlike 100% DL-sample in the intra-HNCACB spectrum.

We now explained this in the text (section 2.2) and the new Figure 2.

>More detailed information would be appreciated about amino acid type based >on variation of Cb peak intensity. For what type of amino acids? how different >the intensity compares to the normal one?

Response:  We added a new section (2.2) with a new Figure 2 and discuss the variation of Cβ peak intensities. We compared the differences between 20% and 100% samples (data not shown). However, different S/N in the different spectra from different samples and influences from the aerobic condition during protein expression for some amino acid types make it challenging to perform quantitative analysis with meaningful data. Therefore, we used Cα peaks as the internal references, which seems to be more reliable to estimate the Cα-Cβ bond connections in the fractional 13C-labeled sample. This analysis is presented in Figure 2E.